# Ultra-large library screening with an evolutionary algorithm in Rosetta (REvoLd)
Paul Eisenhuth [1,2] ✉, Fabian Liessmann [1], Rocco Moretti [3] & Jens Meiler[1,2,3,4]

Ultra-large make-on-demand compound libraries now contain billions of readily available compounds. This represents a golden opportunity for in-silico drug discovery. One challenge, however, is the time and computational cost of an exhaustive screen of such large libraries when receptor flexibility is taken into account. We propose an evolutionary algorithm to search combinatorial make-on-demand chemical space efficiently without enumerating all molecules. We exploit the feature of make-on-demand compound libraries, namely that they are constructed from lists of substrates and chemical reactions. Our algorithm RosettaEvolutionaryLigand (REvoLd) explores the vast search space of combinatorial libraries for protein-ligand docking with full ligand and receptor flexibility through RosettaLigand. A benchmark of REvoLd on five drug targets showed improvements in hit rates by factors between 869 and 1622 compared to random selections. REvoLd is available as an application within the Rosetta software suite (https://docs.rosettacommons.org/docs/latest/revold). This work formulates an evolutionary algorithm for optimization and exploration of ultra-large make-on-demand libraries. We demonstrate that our approach results in strong and stable enrichment, offering the most efficient algorithm for drug discovery in ultra-large chemical space to date.

Drug discovery is a complex and time-consuming process. A campaign typically starts with target selection, followed by hit identification. Hit identification is usually done through screening experiments. While other factors play a role, chances of success increase with the number of tested compounds. However, acquiring a large number of molecules is expensive and testing them in bulk requires specialized infrastructure[1]. One widely adopted solution is virtual high-throughput screening (vHTS), where molecules are pre-screened on computers and filtered for predicted activity[2]. One of the most significant challenges limiting vHTS is the size of the chemical space, which is estimated to contain up to $10^{60}$ possible drug-like molecules[3]. In addition to the lack of computational capacity to store and screen such a large number of molecules, defining a chemical space that is drug/lead-like and synthetically accessible presents a hindrance[4–6]. A wide range of technologies has been developed to generate molecules tailored to specific areas of interest. However, many computational approaches are never thoroughly tested as the barrier for synthesis is too high and, thus, compounds are unavailable for in-vitro testing[6–11]. Make-on-demand combinatorial libraries can overcome this challenge if vHTS algorithms

can be tailored to sample this focused but still prodigious chemical space. These libraries combine simple building blocks through robust reactions to form billions of readily and economically available molecules. In the best-case scenario, they allow the confirmation of bioactive hit molecules from in-silico prediction through in-vitro evaluation within a few weeks[12–14]. However, the increased make-on-demand chemical space is not only an opportunity but also a challenge. A small number of vHTS campaigns have been conducted on molecule libraries exceeding a hundred million compounds, even fewer exceeding billions, and they all required substantial computational resources[13,15–17]. Additionally, most of the computational time in such campaigns is spent on molecules of no further interest for the following steps of the drug discovery campaign due to low hit rates.

The majority of these vHTS campaigns utilize rigid docking, as it tremendously decreases the computational demands compared to flexible docking. However, this introduces potential error sources, as rigid docking might not be able to sample favorable protein–ligand structures[18]. This is in line with previous findings where the introduction of both protein and ligand flexibility increased success rates notably[19–22]. Throughout this study,

[1]Institute for Drug Discovery, Leipzig University, Leipzig, Germany. [2]Center for Scalable Data Analytics and Artificial Intelligence (ScaDS.AI) Dresden/Leipzig, Leipzig University, Leipzig, Germany. [3]Center for Structural Biology, Vanderbilt University, Nashville, TN, USA. [4]Department of Chemistry, Vanderbilt University, Nashville, TN, USA. ✉e-mail: eisenhuth@cs.uni-leipzig.de

we used the RosettaLigand flexible docking protocol[23–25]. It is well-positioned among other available methods and showed strong ranking capabilities during screens of the Enamine REAL space[25–27].

To date, several solutions have been proposed to address this problem. The Deep Docking platform[28,29] utilizes a mixture of conventional docking algorithms and neural networks to screen a subset of the target space and quantitative structure–activity relationship (QSAR) models to evaluate the remaining target space. However, this approach, called active learning, still requires the docking of tens to hundreds of millions of molecules and calculating QSAR descriptors for the whole billion-sized molecular library. A similar idea is used, for example, by Luttens et al.[30], RosettaVS[31], MolPal[32] and HASTEN[33].

Another promising solution is V-SYNTHES[34,35]. Instead of docking the final molecules, V-SYNTHES starts with docking of single fragments, picking the most promising ones, and iteratively adding more fragments to the growing scaffolds until final molecules are built. SpaceDock follows the same concept, but is not limited to commercially available combinatorial libraries[36]. Chemical Space Docking is essentially the same approach as V-SYNTHES and SpaceDock, but is a general instruction instead of being a ready-to-use software[37]. A similar approach, called Targeted Exploration[38], filters the synthons of Enamine's REAL space[39] for similarity to known binders. The most promising synthons are used to enumerate ligand libraries. Search on chemical space near functional molecules requires previous structural knowledge of the molecules, which is not always available. To create such a space, millions of computational docking procedures have to be performed. Other active learning algorithms are SpaceHASTEN[40] and Thompson Sampling[41].

Recently[42], published an evolutionary algorithm called Galileo to optimize molecules in chemical combinatorial space. The algorithm is not tailored towards a specific optimization goal or chemical space, but accepts any function that assigns a score to a molecule and treats reaction rules as general as possible. The algorithm was tested for a similarity search and optimization of pharmacophores, although with mixed success. A total of five million fitness calculations in the context of structure-based drug design makes expansive docking models unfeasible. Another approach using an evolutionary (or genetic) algorithm is SpaceGA[43]. It utilizes established mutation and crossover rules and maps the resulting molecules back to the combinatorial chemical space through similarity search with SpaceLight[44]. Both algorithms showed promising performance. This is in line with recent analysis on the potential of genetic algorithms, showing that their capabilities are on par with modern deep learning methods[45,46]. Evolutionary algorithms have been used for decades in computer-aided drug discovery (CADD) and were implemented by multiple research groups[7,47–50]. They were all highly successful in finding and optimizing promising compounds, but shared one common drawback—synthetic accessibility. One recently published evolutionary algorithm even puts most of its research effort into assuring easy synthesizability[51].

Building on these findings, we propose RosettaEvolutionaryLigand or short REvoLd: An evolutionary algorithm optimizing entire molecules from the Enamine REAL space[39]. It reveals promising compounds with just a few thousand docking calculations, continues to discover new scaffolds if run multiple times, and enforces high synthetic accessibility. Furthermore, REvoLd's enrichment capabilities seem to be independent of the size of the space searched.

## Results
### Hyperparameter and protocol optimization
REvoLd is very flexible and has endless potential protocols (e.g., combinations of selectors, reproduction steps, and several global parameters). To allow for quick testing and optimization of the evolutionary protocol, we created a subset of the Enamine REAL Space consisting of one million scored molecules. This is described in further detail in the section "Pre-docked benchmark". An iterative approach was used to test different combinations of selection and reproduction mechanics and run parameters. Initially, we selected parameters to bias towards the fittest individuals,

allowing only them to mutate and reproduce. This setting proved to converge very fast towards the hit molecules. Its downside was limited exploration of the target space. We introduced several changes to our protocol to offset that effect. First, we increased the number of crossovers between fit molecules to enforce more variance and recombination between well-suited ligands. Second, we added an additional mutation step, which switches single fragments to low-similarity alternatives. This keeps well-performing parts of promising molecules intact but enforces huge changes on small parts of it. And third, another mutation step, which only changes the reaction of a molecule and searches for similar fragments within the new reaction group. This opens larger parts of the combinatorial space for screening. These changes increased the number and diversity of virtual hits tremendously. As a last change, we introduced a second round of crossover and mutation, excluding the fittest molecules, thus allowing worse-scoring ligands to improve and carry their molecular information forward. These changes reliably improved hit rates. Details on tested parameter combinations as well as their hit rates can be found in Supplementary Note 1.

Regarding the size of the random start population, we found that 200 initially created ligands offer enough variety to start the optimization process. More initial ligands might increase the chance of discovering good binders immediately, but greatly increase run-time costs. Fewer initial molecules, on the other hand, have less chance to capture promising structural elements. Next, we tested how many individuals should be allowed to advance to the next generation and found 50 to perform best. Larger populations carry more noise through the generations, which decreases the effectiveness of all reproduction steps, but smaller populations are too homogeneous and therefore hinder exploration of chemical space. Lastly, we found 30 generations of optimization to strike a good balance between convergence and exploration. Good solutions are usually unveiled after 15 generations, but only after 30 generations a flattening of discovery rates has been observed. The algorithm never fully converges and continues to discover well-scored molecules even after 400 generations, but the hit rates become smaller and smaller. Therefore, we advise multiple independent runs instead. The random starting population seeds different paths, which yield different high-scoring motifs. Since each run unveils new promising molecules, the exact number of required runs is solely depending on desired amount of hits.

Additionally, we observed that our algorithm was unable to discover the lowest-scoring molecule of the benchmark test subset of one million molecules. This might be related to the mentioned intended ruggedness of our scoring landscape, which traps runs at local minima. On the other side, it is not uncommon to observe only close-to-optimal solutions from meta-heuristic optimization algorithms like evolutionary algorithms. Considering REvoLd's purpose, we postulate that this is not a flaw, since structure-based CADD campaigns almost never want to discover the single best-scoring compound, but many promising compounds which will enrich hit rates in in-vitro experiments.

### Benchmark under realistic conditions
Based on the first results and selected parameters, we moved on to more realistic benchmarking conditions, utilizing our largest available Enamine REAL space[39] with over 20 billion molecules at that time. Details on the five used drug targets and data collection can be found in the section "Drug target data collection". Twenty runs of REvoLd were conducted against each target, docking between 49,000 and 76,000 unique molecules in total per target. The difference in sampled molecules per target is due to the stochastic nature of evolutionary optimization, as one run might produce more duplicates than another. This includes all docked molecules during the evolutionary optimization, not just the last generation. Figure 1 shows the development of scores in a selected run for each target. All runs successfully reported molecules with hit-like scores. Due to the size of the defined chemical space searched and the stochastic nature of our protocol, there was only a small overlap between the runs. We found that between 1.5% and 3% of tested compounds have a Tanimoto similarity of 1.0 to another compound tested against the same target. These duplicates were removed for further analysis. The performances in Fig. 1 show that four out of five runs

**Fig. 1 | Score development during single REvoLd runs.** Score distribution statistics from single selected runs are shown for each protein target. Solid lines show the best score found up to a certain generation. The dashed line shows the 10th best score, the dotted the 100th best score and dashed-dotted the median. The runs that reported the best-scoring molecule for each target were selected as examples here. All scores within the gray area are better than the best-scoring known active.

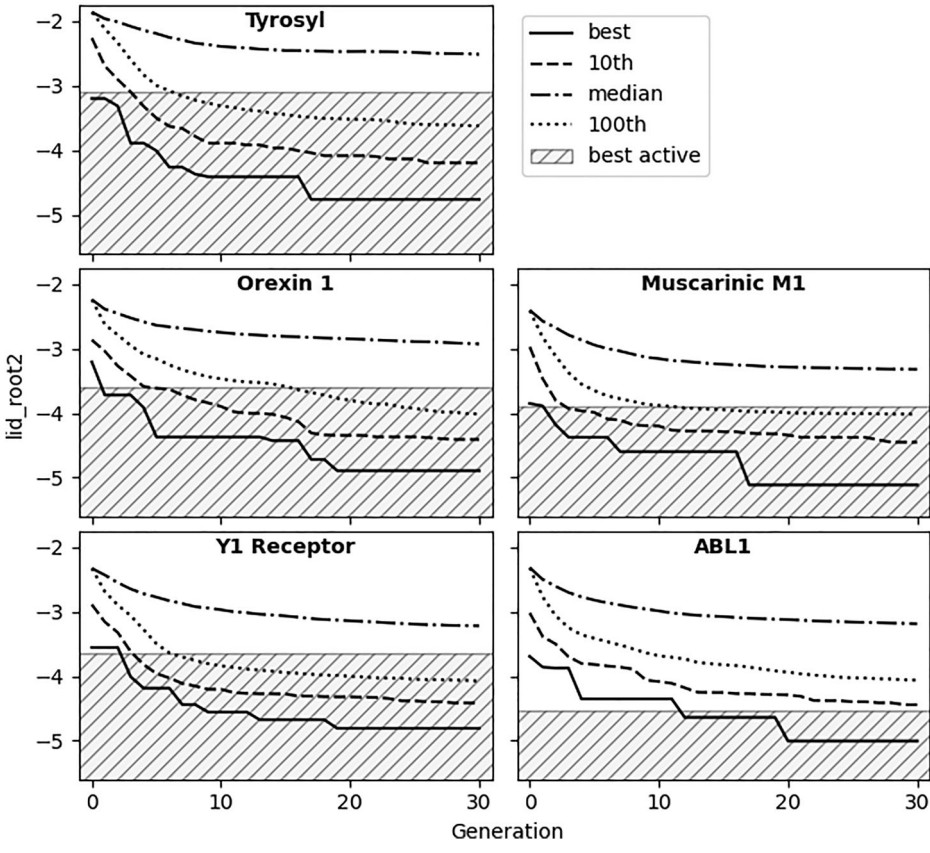

found compounds scoring as good as the best known active within the first 200 randomly sampled molecules. While this seems intuitively unlikely, it highlights a shortcoming of the deployed scoring function. As we discussed before, the distribution of scores for known actives is only slightly more negative than the 100,000 large random sample of Enamine molecules. This means there is a significant overlap between known active and random scores. We found that for the four cases in question, between 21 and 45 molecules out of 100,000 show better scores than the best-known active. This translates to a 4–8% chance of observing at least one molecule with such good scores in a sample of 200 initial candidates. These numbers are in line with the presented performances, as we are showing only one out of 20 runs for each target. Additionally, only one of these runs actually found a better-than-active molecule in generation 0; the rest were just within the general scoring range.

To assess the overall sampling performance, we use enrichment factors. As is often the case for in-silico evaluations, we assume a perfect predictive power for a scoring function at a given threshold and that all compounds scoring better than this threshold will be active in experimental validation. This decouples the performance of a sampling algorithm from its underlying scoring method. The number of hits achieved through a selected sampling method is compared to the number of hits from a random sample. Each molecule in the random sample is selected by first sampling a reaction (weighted by the number of total products in that reaction) and then uniformly sampling one reagent for each available position. This follows the same sampling approach used for the initial starting population and is repeated until 100,000 random molecules are generated. For better comparability, we followed the enrichment factor proposed with V-SYTNHES[34] but added a normalization factor for sample sizes. This is implicitly still the same function as V-SYNTHES as they used samples of the same size:

$$EF_i(x) = \frac{HitRate_i(x)}{HitRate_{rs}(x)} \quad (1)$$

where $EF_i$ is the enrichment factor of method $i$, $x$ defines the score limit, rs is the random sample and

$$HitRate_i(x) = \frac{|\{m \mid m \in M_i, score(m) < x\}|}{|M_i|} \quad (2)$$

where $M_i$ is the set of all molecules sampled with method $i$. The enrichment rate can be interpreted as how many more hits can be expected from a constant sample size or how many fewer molecules need to be sampled for the same number of hits. We propose to use this calculation for all the following exploration algorithms for better comparability.

Figure 2 reports absolute hit numbers of REvoLd and the random sample for different hit limits, as well as the normalized enrichment of REvoLd over the random sample. REvoLd achieves maximum enrichments up to 869 and 1622 between all five targets, outperforming all currently available algorithms, which enable drug discovery in ultra-large libraries. Additionally, Fig. 2 shows that REvoLd is able to report hundreds of hits for much stricter hit limits than a random sampler. We also want to report the enrichment rates using the score of the best-scoring known active as the hit limit, following widespread practice. REvoLd achieves enrichments in four cases between 200 and 532. No such enrichment can be reported for the ABL1 kinase, since the random sample did not include a single molecule scoring better than the best-scoring known active, but REvoLd unveiled 99 such compounds. Using the best scoring known active as a hit limit can be important because scoring functions tend to overrate molecules showing certain artificially favored structural features. At the same time, this limit alone falls short in comparing sampling strategies' capabilities to optimize into local minima.

Furthermore, we observed the same convergence behavior as in the section "Hyperparameter and protocol optimization". No run stopped sampling new molecules until the final generation, but the number of new unique structures per generation started to become relatively low. We could also observe a linear correlation between the number of tested unique

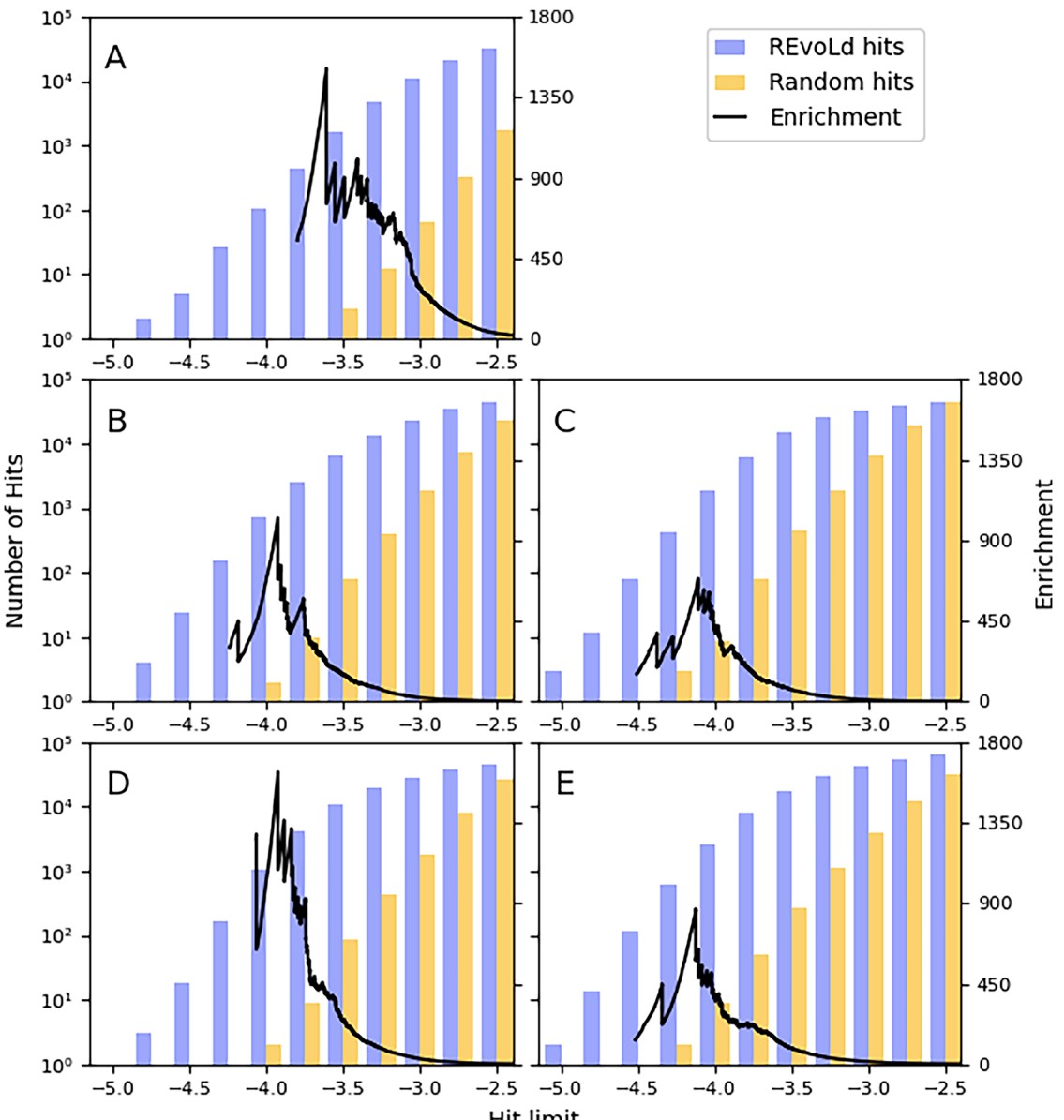

**Fig. 2 | Total number of hits and enrichments.** We tested REvoLd against five different protein targets, namely Tyrosyl (**A**), Orexin 1 (**B**), Muscarinic M1 (**C**), Y1 Receptor (**D**) and ABL1 (**E**). We combined the results of 20 independent REvoLd runs against each target into single hit lists and removed duplicates with a Tanimoto similarity of 1.0. A molecule is to be considered a hit if its score is below a given hit limit. The total number of hits from REvoLd for various hit limits is shown (purple) together with the number of hits from a random sample (orange) on a logarithmic scale. REvoLd tested in total between 49,000 and 76,000 molecules; the random samples always consisted of 100,000 molecules. Therefore, we additionally plot the normalized enrichment of REvoLd over the random sample for various hit limits.

molecules and the number of unique Bemis–Murcko scaffolds[52]. A more thorough discussion of the molecular diversity can be found in Supplementary Note 2.

## Qualitative analysis of reproduction mechanics

We conducted a qualitative investigation of the effects of mutation and crossover on a few examples using REvoLd's reproduction functionalities. We found that the in-silico functionalities effectively mimic the alterations to molecules typically conducted by medicinal chemists. Mutation enabled the introduction of small local changes, such as increased flexibility of certain parts of the molecules or alteration of their geometry through changing the attachment atom of a ring. Crossover, on the other hand, recombined promising motifs into new molecules, effectively transferring and combining knowledge from two separate ligands. This resemblance of medicinal chemistry was never intentionally developed or encoded but emerges naturally from evolutionary optimization paradigms. Figure 3

illustrates both operations for one example, including the change of fitness scores. The observed worsening of scores between C to D and G to H, respectively, also highlights the importance of selective pressure to allow temporally worse scores for new molecules. Both shown mutation steps introduce disadvantageous changes, but the resulting molecules recombine into the best-found solution.

## Runtime analysis

All runs were conducted on Leipzig University's high-performance parallel computing cluster equipped with AMD EPYC 7713@2.0 GHz–Turbo 3.7 GHz processors. We tested REvoLd runs with 20, 40, 60, and 100 cores per run. Only the first core, acting as the control and distribution core, loaded the entire Enamine REAL space database using around 23 GB of memory for the current library size. Most of the memory is taken up by representing each fragment with an RDKit molecule object. All other cores were used for docking alone and required less than 4 GB, mainly depending

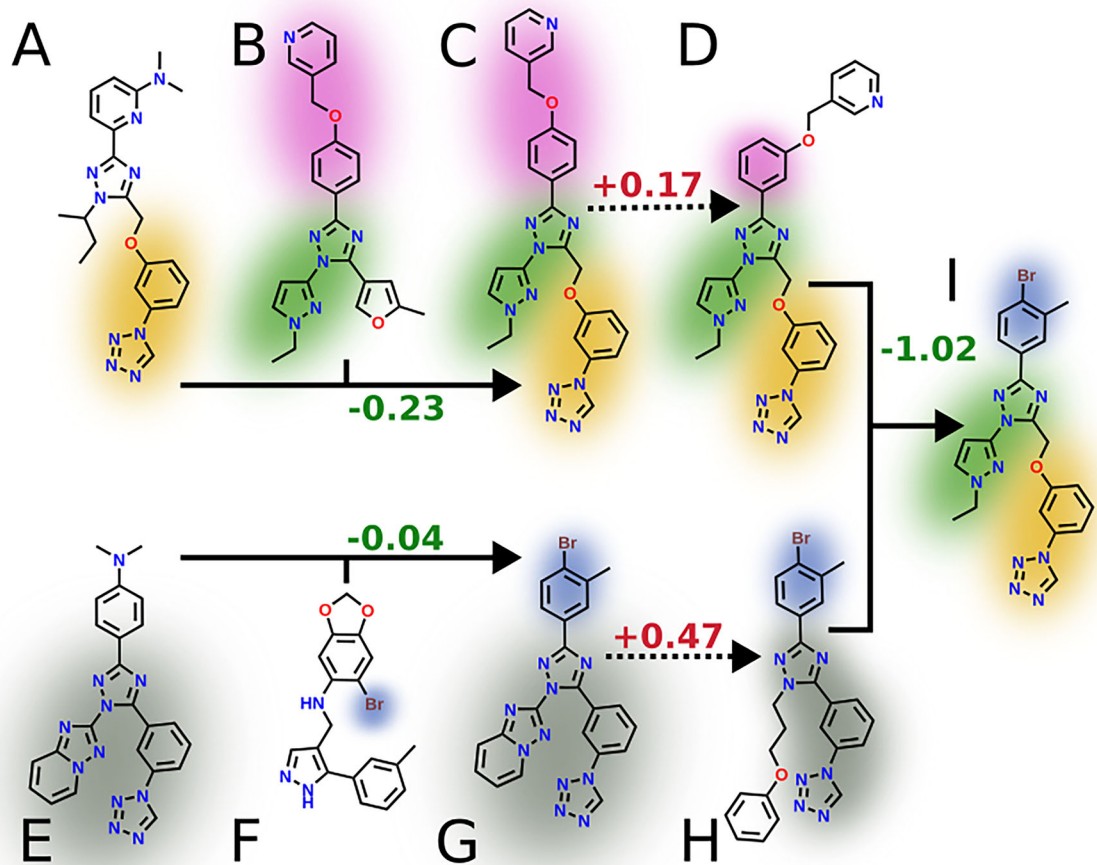

**Fig. 3 | Example of a molecular family tree.** The best-scoring molecule for ABL1 is shown to exemplify mutation and crossover. The color highlights do not show synthons, but reappearing structural motifs. **A** and **B** recombined through crossover into **C**, which mutated to (**D**). Here, **A** includes a tetrazole ring, a common motif in many FDA-approved drugs. Both **A** and **B** contain a 1,2,4-triazole, representing the basis structure to combine it to (**C**). From **C** to **D** a mutation changes the position of the oxymethylpyridine at the benyzl-group from 4 to 3. In the other route, **E** containing a tetrazole and a 1,2,4-triazole, exchanges moieties with **F**, containing

the substructure 6-Bromo-1,3-benzodioxol. Here, instead of exchanging the exact moiety, a similar synthon in the set is searched, adding a 3-methyl,4-bromo-benzyl group to the offspring (**G**). A mutation derivatizes the [1,2,4]triazolo[1,5-a]pyridine from **G** to **H**'s oxybenzyl-moiety. Finally, both **D** and **H** recombine into **I**, introducing the tetrazole-triazole-system from both **H** and **D** while including the pyrazole from **D** and the 3-methyl,4-bromo-benzyl group from (**H**). All reproduction steps happened in different generations. Positive and negative numbers are observed in unfavorable and favorable score changes.

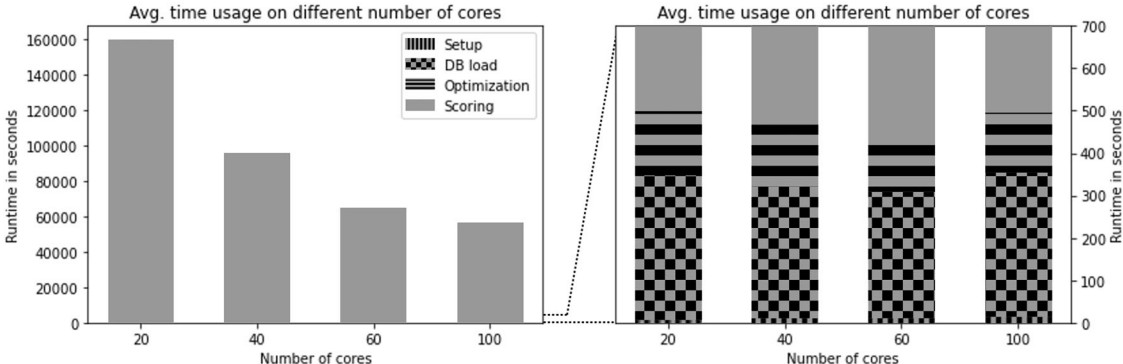

**Fig. 4 | Runtime analysis.** The average runtime of REvoLd on different numbers of CPU cores on the high-performance compute cluster. Over 99% of all computational time is spent on docking (gray). More than half of the remaining time is spent loading and preparing the Enamine REAL space data (checkered). The evolutionary

optimization (horizontal stripes) and setup of Rosetta (vertical stripes) is comparably fast and therefore neglectable. The setup time is barely visible within the figure and takes ~20 s, independent of the number of cores.

on the size of the protein structure. The highly parallel code of REvoLd and its efficient development in C++ caused very fast computation times. A single run with standard settings needs between 24 and 48 h to finish, depending on the number of used CPUs.

As visualized in Fig. 4, more than 99% of the time was spent on docking molecules as REvoLd causes close to no overhead, enabling it to use fully flexible protein and ligand models. Within the time spent on computations outside of docking, most of it is caused by loading and preparing the

database. The speedups gained through the usage of more CPUs are acceptable. 20–40 cores yield a speedup of 1.679 (optimal 2.0), 40–60 cores yield 1.479 (optimal 1.5). The poor speedup from 60 to 100 (1.152, optimal 1.666) is due to the implementation. The docking of a single molecule can only be executed on a single CPU, and if all molecules for one generation are distributed, a CPU is idle until the next generation is docked. Therefore, CPU numbers should be integer fractions of the number of new individuals in each generation. Since this is subject to randomness, higher CPU numbers have higher chances of being idle and therefore reduced speedups.

## Impact of database sizes

We ran REvoLd on different sizes of chemical spaces during its development process. From one million for the hyperparameter optimization, to around 300 million, 1.3 billion, and finally 20.1 billion, following the growth of the Enamine REAL space. The size of the chemical space had a severe impact on our code development and required several restructurings. The biggest consequence of larger chemical spaces is library initialization times and memory requirements. They grew from a few seconds and several hundred megabytes to six minutes and 23 GB, respectively. However, we found no sign that REvoLd's capability to unveil low-scoring compounds is affected by the database size. We found it helpful to increase the exploration parts of the protocol slightly, but the number of individuals and generations remained constant. That means the number of required docking runs stays constant, even if the library size increases dramatically. It should be noted, though, that REvoLd was optimized for efficient runtimes whilst neglecting database preparation times and memory requirements. With the continued growth of combinatorial spaces, this would need reconsideration to adapt to larger quantities of reagents.

It should be noted, though, that REvoLd is not suited to conduct exhaustive screens. The randomness inherent in its protocol makes it unfeasible to sample and dock every molecule. As we observed only very limited overlap between runs, we assume that every run optimizes within an independent subspace of the available chemical library and that 20 runs are not enough to cover all subspaces. If more resources are available and larger numbers of hits or more diverse molecules are required, REvoLd runs should be repeated until the desired criteria are reached. Although we expect larger libraries to require tremendously more runs to cover all subspaces, the number of runs to unveil a set number of hits should remain independent of the library size, making REvoLd very scalable to diverse requirements and limitations.

## Discussion

With the increased availability of three-dimensional protein structures and easily accessible chemical spaces expanding into billions of molecules, structure-based drug discovery and ultra-large library screening are becoming more and more important[12,14,35,53]. It is therefore crucial to improve the available protein-ligand-docking technologies to handle compound libraries of these sizes. Besides reducing the required computational time for available tools, it is also beneficial to reduce the number of required dockings. We have shown that REvoLd is a promising tool to help in that regard. Its evolutionary optimization has proven to be reliable and efficient. We were able to create highly enriched datasets for five different protein targets with only a minimal overhead of computational time. We were additionally able to observe optimization behavior comparable to medicinal chemistry practices.

REvoLd is one of several approaches that try to solve the problem of the size of chemical spaces with an efficient sampling algorithm, like V-SYNTHES and Galileo[34,42]. V-SYNTHES uses a greedy heuristic (meaning it selects fragments only based on their isolated scores and thereby potentially missing combined molecules which exceed the scores of their building blocks), Galileo is another evolutionary algorithm. While neither greedy nor evolutionary is inherently better than the other, the implementations of all algorithms show different performances. Galileo is more of a proof of concept and is not combined with a docking protocol, but accepts any score as fitness and relies on external protocols. Additionally, the

authors reported that they were not able to generate good results for all their targets. REvoLd, on the other hand, showed convergence in all test cases. V-SYNTHES is available as a ready-to-use software and reported good results on all benchmarks, just as REvoLd. However, REvoLd shows greater enrichment (between 869 and 1622 for REvoLd and 250 to 460 for V-SYNTHES, respectively) and requires overall fewer docking runs to achieve these results (between 49,000 and 76,000 for REvoLd and 500,000–1,000,000 for V-SYNTHES). It should also be mentioned that V-SYNTHES requires additional docking of intermediate libraries and single fragments, causing a computational overhead of 20–35%. Additionally, we observed that REvoLd did not require more docking runs for larger combinatorial databases, whereas V-SYNTHES reported a linear complexity relation with the number of used fragments. This relationship exists within REvoLd as well, but only for memory requirements and preprocessing. It should be noted, though, that V-SYNTHES reported success in follow-up in-vitro experiments, while REvoLd is so far only benchmarked in-silico.

Alternative methods like Deep Docking and RosettaVS propose a mixture of structure-based docking protocols and QSAR machine learning instead of sampling-based optimization[28,29,31]. They dock subsets of the whole chemical space, generate molecular fingerprints for all molecules and train a QSAR model on the in-silico docking scores. This gives them an advantage over methods like REvoLd, because they consider information for all molecules and are therefore better suited to discover globally best hit candidates. But this is also a huge disadvantage, because they still require analyzing every molecule, which continues to become more expensive as chemical space grows. Additionally, both methods require several million docking runs and therefore, fall short of REvoLd and V-SYNTHES. A similar number of docking runs is required for Targeted Exploration, which uses prior knowledge about binding fragments to guide the molecule selection[38]. However, the dependence on prior knowledge is also a big limitation. Together with the required number of dockings, Targeted Exploration becomes less optimal as a universal solution for structure-based drug design in ultra-large chemical space.

REvoLd is well positioned among these methods and provides with its sampling capabilities a promising solution for the problem of ultra-large library screens in general. With that, two main topics require further analysis. The biggest barrier for REvoLd right now is the runtime of the employed docking protocol. Reducing the time spent docking will increase REvoLd's speed tremendously, as over 99% of the runtime is used for docking. Machine learning based docking tools like DiffDock, EquiBind and DeepDock are promising candidates to lift these restrictions[54–56]. The other important topic is the reliability of our deployed scoring function when transferring REvoLd's in-silico results to in-vitro testing. Future experiments will need to investigate how prone REvoLd is to overfitting the fitness function. This can happen if a scoring function overestimates certain interactions or substructures in a molecule compared to actual in-vitro findings. Overfitting becomes more important with increasing library size[17].

## Conclusion

REvoLd is a promising algorithm tackling the problem of searching for potential ligands in combinatorial libraries spanning billions of compounds. Through advanced evolutionary optimization, it enables expensive docking methods for such a task. Our benchmarks have shown promising results, and REvoLd reliably outperformed a random sampler over five different targets. Unlike its competing algorithms, REvoLd is capable of discovering molecules with high predicted affinity with the same number of protein-ligand dockings, even in increasingly larger libraries. To date, it is the only complete pipeline to generate in-silico enriched, target-specific compound lists out of combinatorial chemical libraries without the need to dock several hundreds of thousands of molecules or even more. The availability of compounds through industrial make-on-demand services allows easy experimental validation of suggested hits and completely negates the need for synthetic accessibility scores. Future research will need to focus on improving the currently applied docking protocol. Speeding up the docking

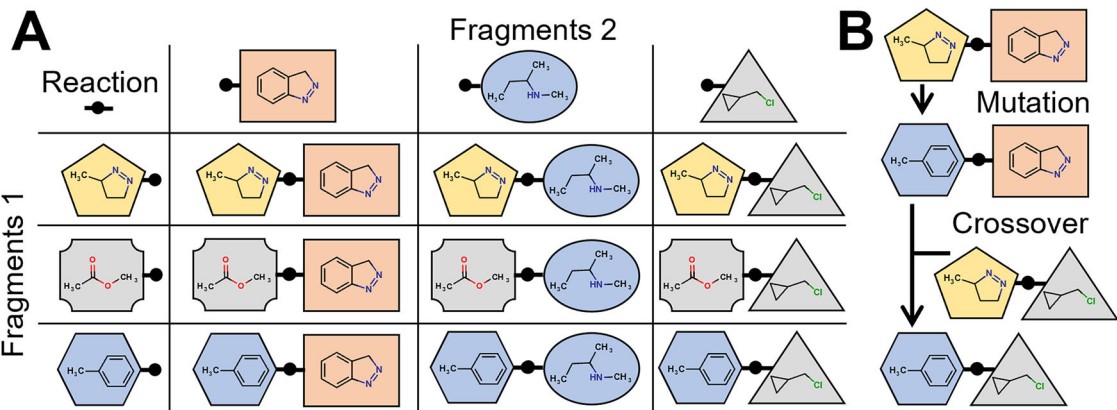

**Fig. 5 | Example of fragment combinations. A** Fragments are defined with attachment points or reaction handles. These are used as hinges to freely combine multiple fragments from predefined lists. For example, the yellow pentagon can be combined with all three fragments in list 2 to form three different products. The product space grows quadratically as both fragment lists grow linearly. **B** An example of mutation and crossover. The mutation aims to replace the fragment from list 1. A local similarity search through all those fragments yielded the blue hexagon. Next, a crossover between the new molecule and another one happens. Both parents contribute one fragment each and create a new molecule that resembles both its predecessors.

calculations will have a significant impact on REvoLd's runtime performance. Additionally, while we found that *lid_root2* can be used to distinguish actives from random samples, further experiments need to investigate the predictive power of RosettaLigand and our normalization approach.

## Methods
Evolutionary algorithms mimic Darwinian evolution[57]. Starting from a random population, individuals are altered, and selective pressure is applied through a fitness function. In each iteration, individuals that are discarded due to their fitness will be replaced by new individuals created through reproduction. In CADD, evolutionary algorithms often utilize docking scores as fitness functions to minimize the free energy between the protein target and the ligand. Reproduction typically consists of mutation and crossover, where mutation alters small parts of the current molecule, like removing or adding a single atom or functional group. Crossover, on the other hand, recombines two or more promising solutions, for example, by cutting two molecules in half and recombining their fragments[58]. Our proposed algorithm sticks to this paradigm[47]. REvoLd allows for all parameters to be changed and adapted by the user. Several selection and reproduction methods can be freely combined to form customized evolutionary protocols.

### Combinatorial libraries
While our reproduction functions follow the core ideas of evolutionary algorithms, their main novelty is their strict limitation to the chemical space defined by the make-on-demand library. Examples of such libraries are Enamine REAL space, Otava CHEMriya space, and WuXi LabNetwork GalaXi space[39,59,60]. While they all differ in size and include different molecules, they all define chemical reactions and sets of fragments that can be combined through these reactions. This causes an exponentially larger chemical space of billions of molecules defined through a few hundred reactions and hundreds of thousands or at most single-digit million fragments[12,61]. These definitions are exploited by our reproduction functions. Each individual in our algorithm represents a single molecule. It is defined through a reaction and a list of fragments used for that specific reaction (Fig. 5). Although we are showing only examples with two-component reactions here to simplify presentation, it should be noted that REvoLd can process reactions of all sizes as long as at least two components are involved.

### Pre-docked benchmark
We generated a test set of reactions to analyze how well REvoLd can sample in a chemical space with known global minima and to optimize its

algorithmic parameters efficiently. Our test set was produced through four two-component reactions and 500 fragments per reagent position. First, we reduced the number of fragments per reaction and position by selecting a random fragment and adding it to an empty list. Iteratively, all remaining fragments for this specific position were calculated a fingerprint-based Tanimoto similarity to all fragments in the list and the one with the lowest average similarity got added to the list as well. This is repeated until 500 fragments are added. The entire process is done for all positions and reactions available. Next, we selected a random reaction to start our final set and again iteratively compared the average similarity between all fragments from selected and unselected reactions and added the reaction with the lowest similarity to the selection until it contained four reactions. This was done in order to create a rigid similarity landscape, which is usually harder to navigate and optimize within, and to provide a high diversity of possible molecules. All one million molecules were docked with RosettaLigand against the human dopamine D3 receptor (PDB ID: 3PBL,[62]) to have their scores readily available, and to allow us to assess how close the algorithm gets to the global minimal score. We set the best-scoring 1000 molecules (0.1%) from this set as virtual hits. The selection of a rigid similarity landscape as well as optimizing only for a single target is potentially suboptimal for protocol optimization, but we did not observe a deterioration of hit rates when switching to the later explained, more realistic benchmark.

### Drug target data collection
For a benchmark closer to realistic conditions, we selected five established drug targets and sampled within the largest available Enamine REAL space[39] with over 20 billion molecules at that time. The reaction SMARTS and reagent SMILES were obtained directly from Enamine under a non-disclosure agreement in January 2022. We assembled a benchmark-set of five proteins as drug targets. All are well-researched with high-throughput screening (HTS) data available and high-quality crystal structures deposited in the protein databank (PDB), namely the G protein-coupled receptor (GPCR) orexin receptor type 1 OX1 (PDB: 4ZJC[63]), the GPCR muscarinic acetylcholine receptor M1 (PDB: 5CXV[64]), the DNA repair enzyme tyrosyl DNA-phosphodiesterase 1 TDP1 (PDB: 6MYZ[65]), the GPCR neuropeptide Y Y1 receptor (PDB: 5ZBQ[66]), and last, the tyrosine-protein kinase ABL1 (PDB: 2HZI[67]). The corresponding HTS data is taken from curated lists of actives from PubChem[68], with the exception of the ABL1 kinase. Its screening data was provided through the directory of useful decoys enhanced (DUD-E), which curated ChEMBL entries[67,69]. We selected the curated PubChem HTS data to ensure high reliability of reported actives, but only four of the eight reported drug targets had available high-quality protein–ligand structures deposited in the PDB. The ABL1 kinase was

**Fig. 6 | General overview of REvoLd's optimization cycle.** Starting from a random population, fitness scores are calculated through ligand docking and solutions are discarded through selective pressure. Alterations to molecules are introduced through offspring factories iteratively. The cycle is continued until a set number of generations is reached.

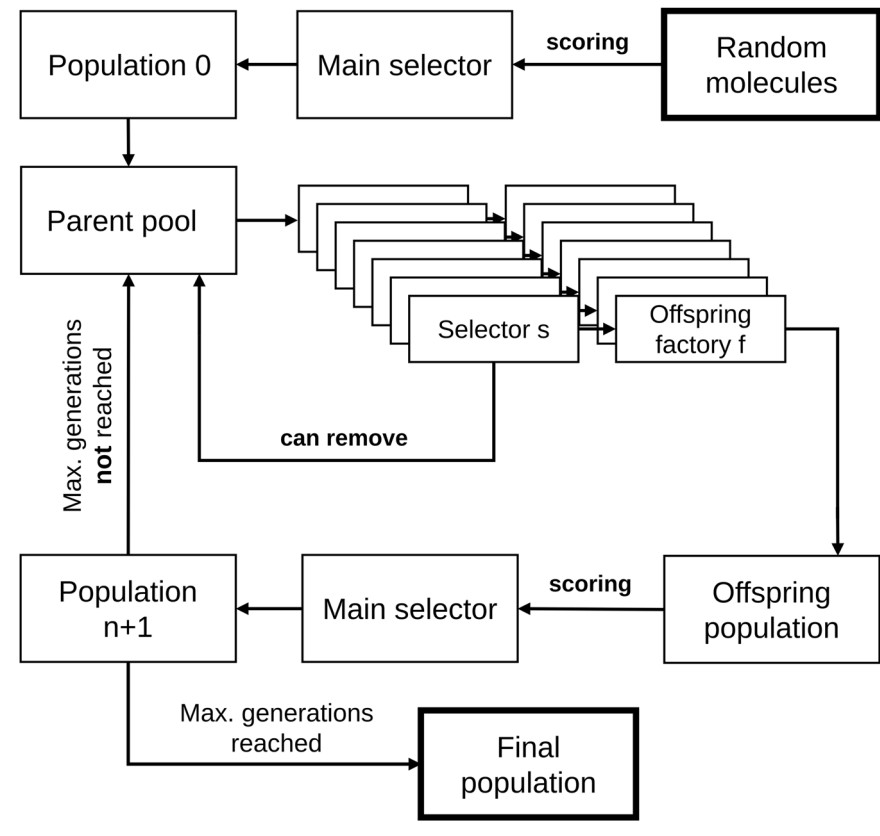

randomly selected from all DUD-E kinases to increase the diversity of protein classes. Through this, we cover a diverse selection of different valuable drug targets and a broad bandwidth of small-molecule ligands, making them a good test case for the benchmark. Furthermore, they represent a mixture of soluble and membrane proteins, and especially GPCRs are known for their flexible nature. Between 188 and 801 molecules are utilized as known actives. All structures were prepared following the presteps of the RosettaLigand protocol, with target sites derived from co-crystallized ligands resolved in complex with the protein structures. All known binders from the used HTS data were docked, again following the RosettaLigand protocol, to use their docking scores to ensure that *lid_root2* helps to enrich molecule sets. Additionally, we sampled 100,000 random molecules from the Enamine REAL space and docked them against each target to compare REvoLd sampling efficiency. We found that the known active score distribution is more negative than the random sample, indicating that *lid_root2* indeed enriches sets of molecules. Details can be found in Supplementary Note 3.

## Algorithm overview

Following typical evolutionary algorithms, REvoLd starts with a random population. An overview of the optimization process is given in Fig. 6. Initial molecules, called individuals, are generated through picking a random reaction and picking one random suitable synthon for each of the reaction's positions. The reaction is picked by a weighted random selection. The weight is the number of possible distinct educts of each reaction. Next, each of these random molecules is docked against the target protein following the RosettaLigand protocol, resulting in 150 complexes per molecule[23,24]. Each of these complexes is used to calculate interface energies between ligand and protein. The lowest calculated interface energy is used as a fitness score. The first population is formed through the application of selective pressure through a freely chosen main selector in order to reduce the number of individuals down to the selected maximum size. It should be noted, though, that whilst we are using the terms individual and molecule often together, they are treated differently. Each individual is an entity in the population,

participates in the evolutionary optimization process and represents one distinct molecule. However, several individuals can represent the same molecule, as it can happen that several recombinations of parents occur multiple times. This is treated with extra care and explained in more detail in the section "Score calculation".

An evolutionary optimization cycle follows to successively select fit individuals, e.g., molecules, for reproduction. Every cycle consists of a sequence of reproduction steps. Each step selects individuals from the previous generation for a given reproduction pattern. The selected individuals remain in the pool for further reproduction steps by default, but can be removed if desired. Next, new molecules are docked to calculate their fitness, and the main selector is applied to reduce the new population down to the selected maximum. The main selector can be freely selected from the available selectors described in the section "Selectors". A cycle is finished by checking if the maximum number of generations is reached. If not, another cycle follows; otherwise, the algorithm ends and reports all analyzed molecules.

## Selectors

In the current version of REvoLd, three different selectors were implemented, all of which can be utilized as the main selector and for selecting individuals for reproduction. The simplest selector, called ElitistSelector, simply selects the fittest individuals. The other two selectors, TournamentSelector and RouletteSelector, are non-deterministic and may allow worse-scoring individuals to be selected for reproduction and to advance generations, in order to explore chemical space further and potentially escape local minima[70]. The RouletteSelector takes the relative differences of fitness scores into account by assigning selection chances based on them. If an individual has fitness two times better than another individual, the first will have a two times higher chance of selection. This deviates from the expected exponential correlation between binding free energy (represented by fitness, e.g., docking scores) and binding affinity. We opted for the linear correlation to make the selector softer and increase the chances of low-scoring molecules for selection. TournamentSelector, on the other hand,

solely considers the ranking of individuals. A set number of individuals is selected randomly to participate in a tournament. This number is referred to as tournament size. All individuals are sorted by their fitness and are granted a chance to accept their selection. Therefore, a larger tournament size means a smaller chance to be selected for less fit individuals. In addition, all selectors have the option to either remove selected individuals from the pool or to keep them in the pool to potentially participate in more reproduction steps.

### Reproduction

Reproduction is implemented through three different offspring factories. Each factory needs to be linked to one selector, which provides it with a set of selected individuals. Each factory uses these individuals to produce a set number of offspring. This is applied sequentially following a user-specified evolutionary protocol. The IdentityFactory simply copies the input individuals unchanged into the offspring generation. This is done to preserve and keep already found solutions and allow them to proceed through multiple generations.

The MutatorFactory applies point mutation to all its input individuals. A point mutation is either the change of one fragment to another fitting fragment or a change of reaction. This is expected to introduce small local perturbations to guide individuals into local energy minima. Switching fragments is easy since each reaction provides a list of suitable synthons for all positions, and those can be freely replaced by each other. RDKit's implementation of extended connectivity fingerprints and Tanimoto similarity are used to control the impact of synthon mutation[71–73]. Fingerprints are calculated for all fragments during database loading at program initialization. When a new fragment needs to be selected, all suitable replacements are collected into one list, and their similarity to the original fragment is calculated. Next, cutoff values are applied to ensure a minimal and maximum similarity. The final new fragment is then selected using a weighted random sample with similarity as weights. This can be used to enforce drastic mutations or to allow only fine-grained changes. The mutation of reactions is more challenging. First, a new reaction is selected. Second, new fragments for each position are selected based on maximal Tanimoto similarity to previously used fragments, to limit the changes induced on the molecule.

The last possibility to create offspring is the CrossoverFactory. It randomly combines all its input individuals into parental pairs, which are used to create offspring inheriting traits from them. One parent provides the reaction used for the offspring, and each parent provides a random number of fragments, but always at least one. If both parents use the same reaction, their fragments can be freely combined. If they use different reactions, a local similarity search is used to find the most similar fragments in the reaction used by the offspring, as it is done in the MutatorFactory.

### Score calculation

Each individual represents a molecule that is a potential hit candidate. To estimate its fitness, or score, we use the ligand docking protocol RosettaLigand and its recommended preprocessing steps[23,24]. First, RDKit[71] is used to turn the SMARTS and SMILES representations of the individual's reaction and fragments, respectively, into RDKit reactions and molecules and to run the reaction with the selected fragments. RDKits' implementation of the ETKDG method is further used to calculate a three-dimensional embedding of the molecule and a list of low-energy conformers[74]. It should be noted, though, that as of now, REvoLd is using only the very basic RDKit functionalities for 3D structure generation. For example, there is no special consideration for stereoisomers or different protonation states. The conformers are passed to the RosettaLigand docking protocol to calculate the protein–ligand complex following a Monte Carlo optimization. Briefly, the RosettaLigand protocol consists of an initial placement of the ligand in a user-defined position, which is ideally a previously identified or known binding site. Next, a coarse-grained docking step successively applies rotation, translation, and conformer sampling to guide the ligand into close contact with the protein. This is followed by a high-resolution docking step, which applies only small changes to the ligand and optimizes protein

sidechains. Lastly, a final minimization step optimizes the current solution into a local minimum. This protocol is repeated 150 times, and the lowest (best) reported protein-ligand interface energy is used as the basis of the efficiency score.

As mentioned before, several individuals can represent the same molecule. This is captured by the algorithm. First, it checks if a molecule has been docked in a previous generation and reuses the same score again. If multiple identical but unscored molecules are present within the same population, only one full docking run is conducted, and the same score is used for all of the identical copies. To ensure diversity across a population, we apply a similarity penalty. Starting from the best scoring molecule, for each molecule, a fingerprint is calculated, stored and compared with all stored fingerprints. If the Tanimoto similarity between two fingerprints exceeds 0.95, a penalty of $+0.5$ is applied to the worse-scoring molecule. This is done to prevent a population takeover by a single well-scoring molecule. At the same time, it does allow very good scoring molecules to be presented by 3–4 individuals (as the first one receives no penalty, the second a penalty of 0.5, the third 1.0, and so on), increasing its chance to pass on genes and thereby pulling the entire population towards a more favorable chemical space.

### Ligand efficiency

Since many of RosettaLigand's energy terms depend on the size of the molecule, we observed the well-known bias towards larger ligands in our first REvoLd runs. Molecules were getting larger and larger with every generation, since larger molecules can form more interactions with the target protein and therefore receive better scores. However, this does not accurately reflect experimental findings[75]. To address this issue, we normalized the interface energy and tested four different methods with $n \in \{1, 2, 3, 4\}$:

$$fitness_n(x) = \frac{energy(x)}{\sqrt[n]{heavyatoms(x)}} \qquad (3)$$

where $energy(x)$ is the interface energy between the protein target and ligand $x$ calculated by RosettaLigand and $heavyatoms(x)$ is the number of non-hydrogen atoms in the ligand $x$. Increasing $n$ essentially decreases the penalty for large molecules. We found $n = 2$ to perform best. Detailed results can be found in Supplementary Note 4. This measure is the geometric mean between the predicted binding energy and the ligand efficiency. It strikes a good balance between biasing against too large and too small molecules. We call this score "ligand interface delta square root normalized" or short *lid_root2*. It will be used throughout the rest of the paper whenever we report fitness or docking scores.

### Final protocol

There are a total of seven different reproductions in our final protocol, but protocols can be freely adapted. This protocol was developed through optimizing hit rates, which is explained in the section "Hyperparameter and protocol optimization". Each step is implemented through a pair of selectors and an offspring factory. The steps are applied sequentially, and most of them keep the parent pool unchanged; only one removes the selected parent molecules. The first two steps are intended to cause small refinements on promising molecules from the previous generation. The next two steps potentially use the same molecules, but make sure more impactful changes are applied to enhance exploration of chemical space, where step 3 increases exploration within the same reaction and step 4 ensures different reactions are explored as well. Step 5 moves the best molecules unchanged to the next generation and removes them from the pool of reproduction candidates. This is done to preserve the best solutions for future generations and to allow optimization of less optimal molecules through the final two steps.

1. *Moderate mutations*: A roulette wheel selector selects 15 individuals and a total of 30 new molecules are produced through mutation, with every parent being used twice. Mutations occur twice as often on

fragments instead of reactions. Fragment selection is biased towards high similarity with an enforced minimal Tanimoto similarity of 0.6. Selected parents remain unchanged in the current pool.

2. *Excessive crossover*: A roulette wheel selector selects 15 individuals and a total of 60 new molecules are produced through crossover. Parents are randomly paired to generate one new molecule. This is repeated until enough offspring are generated. Selected parents remain unchanged in the current pool.

3. *Drastic mutations*: Like step 1, but only fragments can be mutated, not reactions. Furthermore, fragment selection is still biased towards higher similarity, but a maximum similarity of 0.25 is used as a hard cutoff.

4. *Reaction mutation*: Like step 1, but only reactions are mutated to guarantee exploration of chemical spaces defined by different reactions.

5. *Identity*: The 15 best molecules from the current pool are passed unchanged to the next generation. They are removed from the current pool afterwards.

6. *Moderate mutations*: Same as step 1, but since the 15 best molecules were removed in step 5, individuals with a worse fitness have a higher chance of being selected.

7. *Excessive crossover*: Same as step 2, but again with a higher chance for worse molecules.

These steps strike a good balance between exploration and exploitation of chemical space. This cycle is repeated until 30 generations are done, and all molecules from all generations are reported. REvoLd saves the best-scoring protein–ligand complexes calculated during docking for each individual. Our final protocol uses a tournament selector as the main selector with tournament size 15 and acceptance chance of 0.75. The initial population size is 200, and the maximum number of individuals to pass between generations is 50. It is important to make again the distinction between individuals and molecules. The reported numbers are static, meaning that every run results in 7400 individuals. But since these can represent duplicate molecules, we observed on average only 2450–3800 actual molecules being docked per run, as reported in "Benchmark under realistic conditions". While a rate of 50–70% duplicated molecules seems very high, we did not experience a deterioration of results. Additionally, as explained in the section "Score calculation", the duplicates do not cause docking overheads and are penalized.

## Data availability
Protein structures are available through the Protein Databank (PDB), and their codes are mentioned in the paper. Known actives are available either through the PubChem repositories associated with[68] or the directory of useful decoys (DUD-E)[69]. Access to the Enamine REAL space can be requested through Enamine Ltd. The REvoLd guide contains a short section to help with the NDA (https://docs.rosettacommons.org/docs/latest/revold). The data that support the findings of this study are available from the corresponding author, but contain protected intellectual property of Enamine Ltd., which was used under license for the current study, and so are not publicly available. Data are, however, available from the authors upon reasonable request and with permission of Enamine Ltd.

## Code availability
All code is available as part of the Rosetta repository. More information, including compilation and run commands, can be found in the REvoLd guide (https://docs.rosettacommons.org/docs/latest/revold). Rosetta is available under a permissive license for academic research. Commercial utilization requires a separate license.

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

## Acknowledgements

Computations for this work were done (in part) using resources of the Leipzig University Computing Center and of the Vanderbilt University Advanced Computing Cluster for Research and Education (ACCRE). The authors would like to thank Iaroslava Kos and Enamine LTD for their support and access to their dataset. The authors acknowledge the financial support by the Federal Ministry of Education and Research of Germany and by Sächsische Staatsministerium für Wissenschaft, Kultur und Tourismus in the program Center of Excellence for AI-research *Center for Scalable Data Analytics and Artificial Intelligence Dresden/Leipzig*, project identification number: ScaDS.AI. P.E.'s position is funded through an award by ScaDS.AI. J.M. is supported by an Alexander-von-Humboldt professorship. This work was further funded by the Deutsche Forschungsgemeinschaft (DFG, German Research Foundation) through SPP2363 (460865652) and through SFB1423 (421152132). P.E. and F.L. received a fellowship from the Max Kade Foundation to support their work in the Meiler laboratory at Vanderbilt University. Work in the Vanderbilt Meiler laboratory is supported through the NIH (R01 DA046138, R01 HL122010, and R01 GM080403). Funded by the Open Access Publication Fund of Leipzig University.

## Author contributions

Conceptualization: Paul Eisenhuth, Jens Meiler. Data curation: Paul Eisenhuth. Formal analysis: Paul Eisenhuth. Funding acquisition: Jens Meiler. Investigation: Paul Eisenhuth. Methodology: Paul Eisenhuth. Resources: Rocco Moretti and Jens Meiler. Supervision: Rocco Moretti and Jens Meiler. Software: Paul Eisenhuth. Validation: Fabian Liessmann and Paul Eisenhuth. Visualization: Paul Eisenhuth. Writing—original draft: Paul Eisenhuth. Writing—review and editing: Paul Eisenhuth and Fabian Liessmann, Rocco Moretti, Jens Meiler.

## Funding

## Competing interests

P.E. and F.L. are founders of AI-Driven Therapeutics GmbH (AI-DT). The company does not own intellectual property, licenses, or rights associated with the presented work. Both P.E. and F.L. were not employed by AI-DT at the time of writing and received no financial or non-financial compensation from AI-DT. All other authors declare no competing interests.
