## [Transparent Peer Review file · Communications Chemistry]

Ultra-Large Library Screening with an Evolutionary Algorithm in Rosetta (REvoLd)

Corresponding Author: Mr Paul Eisenhuth

Version 0:

Reviewer comments:

Reviewer #1

(Remarks to the Author)

The authors developed a novel evolutionary algorithm, REvoLd, for structure-based virtual screening of ultra-large combinatorial library, specifically Enamine REAL Space, and presented computational validation of this method on five protein targets. This approach allows searching for hit compounds in REAL Space of 20 billion compounds while performing the docking of only a small fraction of combinatorial space (<100,000 docked molecules per target).

The presented approach holds a promising potential for exploring ultra-large combinatorial chemical spaces. The performance of the presented method was compared to the previously published V-SYNTHES algorithm, showing similar or higher efficiency. However, the difference in the two approaches is that V-SYNTHES claims to screen the whole combinatorial space in a single run, while a single run of REvoLd visits a fraction of the total space. By fine tuning the number of runs performed by REvoLd, one can achieve a desired balance between available computational resources and coverage of the space.

Here are several major points requiring clarification or additional analysis.

1. The authors show that each individual run of REvoLd retrieves new compounds, which confirms that this method is visiting only a fraction of the REAL Space. 49,000 to 76,000 docking calculations per target reported in this work (lines 363, 485) sometimes are not enough to screen even the sub-space generated from one fragment. Considering that the majority of academic and industry groups can afford to dock a few millions of compounds, how many runs of REvoLd are required to cover the whole combinatorial space of 20 billion Enamine REAL compounds? How does the number of runs required to screen the whole REAL Space will depend on the reagents space? Would it grow linear with the size of the reagent space? Please add the analysis in the text.

2. Authors addressed the problem of chemical diversity of retrieved hits by using various ways of generation of the next population (lines 231-250). Do these steps allow REvoLd to identify new scaffolds when moving from the initial to the final population? Does the diversity of the retrieved compounds depend on the selection of the initial population? Please show the chemical diversity of compounds retrieved by REvoLd from the starting scaffolds in a single run.

3. A. The authors compared the performance of their method to the previously published V-SYNTHES approach (lines 378-382), specifically comparing the enrichment factors that are derived from number of hits recovered by REvoLd versus a docking of a random subset of molecules from REAL Space, thus greatly dependent on the nature of a random subset. However, the description of the random subset from REAL Space and its generation is missing. Which methods were used to select random subset from unenumerated REAL Space? It is important to explain how it was generated and whether it accurately represents the whole REAL Space.

B. How exactly the initial population of 200 molecules is selected? Is it the same as generation of 100,000 random subset? How much the resulting number and quality of hits may depend on this initial random selection in several test runs? Please add this information to the Methods section.

4. The REAL Space consists of molecules generated from two-component and three-component reactions. All the examples presented in the paper are from two-component reactions. Were three-component reactions screened in this work? Considering that the majority of 20 billion REAL compounds come from three-component reactions, are there any modifications required to the protocol when switching from two-component to three-component subspace? Please clarify this in the text.

5. Figure 3 shows that for four out of five proteins, there are virtual hits in the 0th generation that are as good as or, in the case of Tyrosyl receptor, even better than the best scoring known active. Since the molecules in 0th generation are random molecules of a miniscule subset of Enamine REAL (<0.000001%, 200 molecules out of 22 billion REAL Space), the probability of finding hits with such high scores should be close to 0, and definitely shouldn't happen in 4 out of 5 examples.

- A figure with docking poses of best scoring known actives vs best hits from initial and final generation needs to be added.
- Line 29: the hyperlink leads to "Authentication is required" page. How can the reviewer access it?
 - In description to Figure 1, "grows exponentially" should be replaced by "grows polynomially" for the general case of a chemical space constituted by N-component reactions, or quadratically for the discussed example of the two-component chemical reaction.
 - Lines 159-162 – for RouletteSelector, linear correlation between fitness score and selection chance seems to be too lax. Chosen fitness score is based on docking score, which is meant to provide measure of free energy of binding. The relation between free energy and probability is expected to be exponential, not linear. Please clarify in the text.
 - A. In the overview of REvoLd given in Figure 2, evolution of a population at n-th iteration is shown to be performed through parallel application of selector-offspring factory paired operators, where the operators are applied only to the input set of molecules. However, the description given in the section 2.6 (Final Protocol) implies that some of the operators are applied sequentially one after another, and some modify the pool of parent molecules. Please clarify in the text and make appropriate changes to Figure 2.
B. It is unclear what the "main selector" is in the final implementation. Is it selection of top N molecules by ligand efficiency? Please clarify in the text.
C. It is unclear how many children are spawned at each iteration and how the n+1-th iteration pool is created. Please add this information to the paper. Using the numbers provided in the section 2.6 "Final Protocol", $30+60+30+30+15+30+60 = 255$ molecules are produced by offspring factories at each iteration. This number seems to be constant between generations, then the total number of docked molecules should be $200+255*30=7850$ per run. On line 362 it is mentioned that 20 runs per target were performed, which should result in 157,000 docked molecules. The reported number of docked molecules for each target is given on line 363 and is stated to be between 49000 and 76000 unique molecules. Lines 363-365 state "The difference in sampled molecules per target is due to the stochastic nature of evolutionary optimization, as one run might produce more duplicates than another." Does it mean that around 50%-70% of molecules generated in each run are duplicates? Please clarify how the offspring population is generated and make required alterations to Figure 2 and the text of the Final Protocol. If the above calculations accurately represent the algorithm, please discuss how the duplicates affect the algorithm efficiency.
 - Fig 3, 4, C3: Please adjust the panels for the proteins to be in the same order between the figures for ease of comparison. Make each plot the same size in figures 3 and 4, the empty space can be utilized similarly to C3.
 - Lines 269-276: Dividing docking score by either the number of heavy atoms or their square root are common strategies to estimate ligand efficiency to correct docking score bias towards heavy molecules. "lid_root2" sounds like a name of a variable from source code and creates confusion. Please replace mentions of "lid_root2" in the paper by "ligand efficiency" or a similar term.
 - In Figure 5, it is not clear how molecules E and F were recombined into G. The highlights are inconsistent. From the description to the figure, it seems that in molecule E blue highlight should extend to the phenyl ring. In molecule F, there is no obvious way to fragment the molecule such that it would result in a 3-methyl,4-bromo-benzyl group, and the provided highlight is inconsistent. Please revisit the figure and fix this issue.
 - Line 475: "V-SYNTHES uses a greedy heuristic". After carefully looking into V-SYNTHES paper, there was a section called "Structure-guided selection of fragments". Perhaps change the description of heuristic from greedy to structure-based?
 - Line 494: Chemical Space Docking is similar to V-SYNTHES, but not to Deep Docking. This was already discussed in the introduction (lines 65-82).
 - In appendix, Figure C3 appears without description and seems to be cut off at the bottom.

Reviewer #2

(Remarks to the Author)

Eisenhuth, Liessman, Moretti and Meiler have developed a computational method called REvoLd based on genetic algorithm that allows users to dock nonenumerated chemical spaces using RosettaLigand which, unlike many other docking tools, is a docking methodology that also takes the protein flexibility into account. They developed the protocol by first docking one million compounds (generated from Enamine REAL Space) to the human dopamine D3 receptor and then optimized the various parameters of the REvoLd using this data so that it retrieves the best docking score as quickly as possible. Finally, the method itself was validated using five different well-known drug targets. It was shown that by docking 49,000-76,000 per target, they were able to retrieve significant amounts of compounds with good RosettaLigand docking scores from the nonenumerated chemical space (Enamine REAL Space). It can be estimated from the runtimes provided in the manuscript that the method requires reasonable computing resources and time as screening one target takes about two days with 1000 cores (one screening consists of 20 parallel runs, which each run consuming 100 cores simultaneously).

The research topic of docking of ultra-large chemical spaces is highly relevant in the context of both academic and industrial drug discovery as demonstrated by the various related studies published recently in the scientific literature and this study fits very well in the scope of Nature Communications Chemistry. The main novelty within this study lies in the use of the computationally expensive RosettaLigand software to screen vast nonenumerated chemical spaces, which might open new avenues for virtual screening of targets where the flexibility of the target is crucial.

The manuscript is well-written, logically constructed and easy to read. The high capability of REvoLd to retrieve good compounds from Enamine REAL Space as scored by RosettaLigand is demonstrated using several targets.

Given the speed of the development in the field of screening of ultra-large chemical spaces has been in last couple of years, it is not surprising to find that some of the most recent related work is not cited and short discussion around those should be added to the manuscript. The few comments below do not require additional experiments, so this reviewer finds the manuscript suitable for publication after these minor revisions:

1) While the main strength of RosettaLigand (and the whole justification for the huge computational demand increase) versus faster rigid docking software (such as GOLD or Glide) should be the consideration of the target protein as a flexible object, the manuscript lacks completely the discussion of benefits for considering protein flexibility in the context virtual screening. This might be obvious to the authors, but this should be thus added to the introduction where this would be discussed in detail and authors should add some concrete examples as well (to quote internet slang: "pics or it didn't happen"). This reviewer is guessing that a lot of development of RosettaLigand has been done since the publication of 2009 paper (<https://www.sciencedirect.com/science/article/pii/S0022283608014289>) and there must be some nice examples of the benefit of using RosettaLigand vs. simple rigid docking.

2) Following on that topic: does any of the five targets picked here for benchmarking display any benefits for flexible docking vs rigid docking? This should be discussed around Page 11, line 342. If the targets were not selected because of their flexible nature, the reason why they were selected should be stated clearly. There are plenty targets in DUD-E dataset to choose from so the rationale for picking these specific targets here is interesting to know.

3) Section 4.5: The ligand preparation before docking for molecules constructed from Enamine REAL Space needs further elaboration. How does the REvoLd handle stereoisomerism, tautomerism and different protonation states of the ligands? This is important to describe in detail: for example, the number of different stereoisomers multiply the need for docking calculations as each stereoisomer must be considered as a separate structure to dock. Also, protonation states and tautomers can have a large effect on docking scores and there is no universally accepted approach to deal with them.

4) Regarding 3), authors should also elaborate this part there: "RDKit is further used to calculate a three-dimensional embedding of the molecule 208 and a list of low-energy conformers.". There are various ways on generating conformers in RDKit. If they use the ETKDG method, this should be cited as well (<https://pubs.acs.org/doi/10.1021/acs.jcim.5b00654>)

5) Section 3.2.: The authors should elaborate the generation of diverse one million compounds a bit more as at least this reviewer is struggling to understand how you select a subset of four reactions from a bigger pool of reactions based on diversity selection of building blocks.

6) Regarding the missing references in the introduction:

a. Page #2, Line #71: When discussing methods that mix conventional docking and machine learning methods, the authors should also mention here MolPal (<https://pubs.rsc.org/en/content/articlelanding/2021/sc/d0sc06805e>) and HASTEN (<https://pubs.acs.org/doi/full/10.1021/acs.jcim.3c01239>).

b. Page #3, Line #77: When discussing methods that approach the problem of screening of ultra-large libraries by docking fragments, SpaceDock should be cited as well (<https://pubs.acs.org/doi/10.1021/acscentsci.3c01521>)

c. Page #3, Line #79: there are also methods that use chemical similarity of whole molecules while still navigating in nonenumerated chemical space and they should be also mentioned here: Thompson Sampling (<https://pubs.acs.org/doi/10.1021/acs.jcim.3c01790>), SpaceGA (<https://pubs.acs.org/doi/10.1021/acs.jcim.4c01308>) and SpaceHASTEN (<https://pubs.acs.org/doi/10.1021/acs.jcim.4c01790>).

7) Rosetta requires a separate license for industrial researchers (see <https://els2.comotion.uw.edu/product/rosetta>). The need for this additional license should be explicitly stated in "5.7 Code availability".

Version 1:

Reviewer comments:

Reviewer #1

(Remarks to the Author)

The Authors have fully addressed each comment and question raised during the initial review. I recommend the manuscript for publication.

Reviewer #2

(Remarks to the Author)

All the requested revisions were made, no further comments from my side.

REvoLd manuscript revision reply

Reviewer #1 comments:

The authors developed a novel evolutionary algorithm, REvoLd, for structure-based virtual screening of ultra-large combinatorial library, specifically Enamine REAL Space, and presented computational validation of this method on five protein targets. This approach allows searching for hit compounds in REAL Space of 20 billion compounds while performing the docking of only a small fraction of combinatorial space (<100,000 docked molecules per target).

The presented approach holds a promising potential for exploring ultra-large combinatorial chemical spaces. The performance of the presented method was compared to the previously published V-SYNTHES algorithm, showing similar or higher efficiency. However, the difference in the two approaches is that V-SYNTHES claims to screen the whole combinatorial space in a single run, while a single run of REvoLd visits a fraction of the total space. By fine tuning the number of runs performed by REvoLd, one can achieve a desired balance between available computational resources and coverage of the space.

Here are several major points requiring clarification or additional analysis.

We would like to thank the reviewer for the critical assessment of our work. We truly believe that addressing all comments improved the quality of our manuscript tremendously and hope that we could answer all open questions.

1. The authors show that each individual run of REvoLd retrieves new compounds, which confirms that this method is visiting only a fraction of the REAL Space. 49,000 to 76,000 docking calculations per target reported in this work (lines 363, 485) sometimes are not enough to screen even the sub-space generated from one fragment. Considering that the majority of academic and industry groups can afford to dock a few millions of compounds, how many runs of REvoLd are required to cover the whole combinatorial space of 20 billion Enamine REAL compounds? How does the number of runs required to screen the whole REAL Space will depend on the reagents space? Would it grow linear with the size of the reagent space? Please add the analysis in the text.

This is an interesting question. We deemed the lower amount of dockings enough for benchmarking purposes, but this can certainly be increased for production runs by increasing the number of runs. We added a section starting at line 550 to clarify this topic within the paper: **It should be noted though that REvoLd is not suited to conduct exhaustive screens. The randomness inherent in its protocol makes it unfeasible to sample and dock every molecule. As we observed only very limited overlap between runs, we assume that every run optimizes within an independent subspace of the available chemical library and that 20 runs are not enough to cover all subspaces. If more resources are available and larger numbers of hits or more diverse molecules are required, REvoLd runs should be repeated until the desired criteria are reached. Although we expect larger libraries to require tremendously more runs to cover all subspaces, the number of runs to unveil a set number of hits should remain independent of the library size making REvoLd very scalable to diverse requirements and limitations.**

2. Authors addressed the problem of chemical diversity of retrieved hits by using various ways of generation of the next population (lines 231-250). Do these steps allow REvoLd to identify new scaffolds when moving from the initial to the final population? Does the diversity of the retrieved compounds depend on the selection of the initial population? Please show the chemical diversity of compounds retrieved by REvoLd from the starting scaffolds in a single run.

We thank the reviewer for this excellent remark. We added the following section to line 489: **Furthermore, we observed the same convergence behavior as in 3.2. No run stopped sampling new molecules until the final generation, but the number of new unique structures per generation started to become relatively low. We could also observe a linear correlation between the number of tested unique molecules and the number of unique Bemis-Murcko scaffolds [6]. A more thorough discussion of the molecular diversity can be found in Appendix D.**

Additionally, we added Appendix D to line 758: In section 3.3 we stated that REvoLd reliably samples new molecular scaffolds for all five target proteins. Figure D4 shows on the left side that the number of unique scaffolds and the number of unique molecules are linearly related. This indicates that the deployed sampling approaches are successfully exploring chemical space. We plotted the runs with lowest, median and highest number of unique scaffolds in the final generation from all runs targeting all proteins. There was no observable difference between the targets. Additionally, the right side shows that the number of unique scaffolds increases quickly during the first generation, but starts to slow down around generation 10. This is line with our findings from section 3.2 where we saw a decrease of discovery rates during later generations.

Next, we investigated how much the sampled molecules depend on the run starting population. Therefore, we calculated the maximum Tanimoto similarity to a molecule in the start population for each molecule in all following generations and reported the mean for each run and each generation. The distribution of these mean similarities are shown as box plots. Figure D5 shows that all runs develop away from the starting population but maintain some similarities. Here we show again separate figures for each target due to slightly different observations. The difference between the final set of molecules and the starting population is further highlighted by the four example molecules in the top right corner of figure D5. We selected the run with the lowest diversity between start and final population (which targeted Tyrosyl) and binned both population by molecular size into three groups. The displayed molecules are scaffolds of the best scoring entries from the middle and large bin. We omitted the small size bin, as its entry scaffolds were very fragment-like. To clarify, the displayed similarities are between molecules, only the four examples are scaffolds.

3. A. The authors compared the performance of their method to the previously published V-SYNTHES approach (lines 378-382), specifically comparing the enrichment factors that are derived from number of hits recovered by REvoLd versus a docking of a random subset of molecules from REAL Space, thus greatly dependent on the nature of a random subset. However, the description of the random subset from REAL Space and its generation is missing. Which methods were used to select random subset from unenumerated REAL Space? It is important to explain how it was generated and whether it accurately represents the whole REAL Space.

That is absolutely correct, we apologize for missing this important information. We added a section at line 458: **Each molecule in the random sample is selected by first sampling a reaction (weighted by the number of total products in that reaction) and then uniformly sampling one reagent for each available position. This follows the same sampling approach used for the initial starting population and is repeated until 100,000 random molecules are generated.**

B. How exactly the initial population of 200 molecules is selected? Is it the same as generation of 100,000 random subset? How much the resulting number and quality of hits may depend on this initial random selection in several test runs? Please add this information to the Methods section.

The exact selection for the initial population is explained on line 150 under algorithm overview: **Initial molecules, called individuals, are generated through picking a random reaction and picking one random suitable synthon for each of the reaction's positions. The reaction is picked by a weighted random selection. The weight is the number of possible distinct educts of each reaction.** This is the same approach used for the 100,000 subset. The size of the starting population was discussed in line 371 in the hyperparameter section: **Regarding the size of the random start population, we found that 200 initially created ligands offer enough variety to start the optimization process. More initial ligands might increase the chance to discover good binders immediately, but greatly increases run time costs. Fewer initial molecules on the other hand have less chance to capture promising structural elements.**

4. The REAL Space consists of molecules generated from two-component and three-component reactions. All the examples presented in the paper are from two-component reactions. Were three-component reactions screened in this work? Considering that the majority of 20 billion REAL compounds come from three-component reactions, are there any modifications required to the protocol when switching from two-component to three-component subspace? Please clarify this in the text.

Excellent point. We added a section at line 144 to clarify this: **Although we are showing only examples with two component reactions here to simplify presentation, it should be noted that REvoLd can process reactions of all sizes as long as at least two components are involved.**

5. Figure 3 shows that for four out of five proteins, there are virtual hits in the 0th generation that are as good as or, in the case of Tyrosyl receptor, even better than the best scoring known active. Since the molecules in 0th generation are random molecules of a miniscule subset of Enamine REAL (<0.000001%, 200 molecules out of 22 billion REAL Space), the probability of finding hits with such high scores should be close to 0, and definitely shouldn't happen in 4 out of 5 examples. A figure with docking poses of best scoring known actives vs best hits from initial and final generation needs to be added.

We apologize for the incomplete explanation and the resulting misleading nature of figure 3. We added the following section to line 430: **The performances in figure 3 show that four out of five runs found compounds scoring as good as the best known active within the first 200 randomly sampled molecule. While this seems intuitively unlikely it highlights a shortcoming of the deployed scoring function. As we discussed before, the distribution of scores for known actives is only slightly more negative than the 100,000 large random sample of Enamine molecules. This means there is a significant overlap between known active and random scores. We found that for the four cases in question between 21 and 45 molecules out 100,000 show better scores than the best known active. This translates to a 4%-8% chance of observing at least one molecule with such good scores in a sample of 200 initial candidates. These numbers are in line with the presented performances as we are showing only one out of twenty runs for each target. Additionally, only one of these runs actually found a better-than-active molecule in generation 0, the rest were just within the general scoring range.**

We think this addresses the raised concern much better than a figure of docking poses and therefore did not include it. However, we are of course happy to provide these if the reviewer thinks they are valuable to our manuscript.

6. Line 29: the hyperlink leads to "Authentication is required" page. How can the reviewer access it?

We apologize for this mistake. The included link is to edit the page, we switched to the publicly available access link <https://docs.rosettacommons.org/docs/latest/revold>. This mistake repeated in the data availability and code availability section. All instances are fixed.

7. In description to Figure 1, "grows exponentially" should be replaced by "grows polynomially" for the general case of a chemical space constituted by N-component reactions, or quadratically for the discussed example of the two-component chemical reaction.

We appreciate you spotting this error. We replaced the description with **quadratically** to fit the example.

8. Lines 159-162 – for RouletteSelector, linear correlation between fitness score and selection chance seems to be too lax. Chosen fitness score is based on docking score, which is meant to provide measure of free energy of binding. The relation between free energy and probability is expected to be exponential, not linear. Please clarify in the text.

We agree, this is unintuitive. We have added a section at line 186 to help clarify: **This deviates from the expected exponential correlation between binding free energy (represented by fitness, e.g. docking scores) and binding affinity. We opted for the linear correlation to make the selector softer and increase chances of low scoring molecules for selection.**

9. A. In the overview of REvoLd given in Figure 2, evolution of a population at n-th iteration is shown to be performed through parallel application of selector-offspring factory paired operators, where the operators are applied only to the input set of molecules. However, the description given in the section 2.6 (Final Protocol) implies that some of the operators are applied sequentially one after another, and some modify the pool of parent molecules. Please clarify in the text and make appropriate changes to Figure 2.

We would like to thank the reviewer for bringing this inconsistency to our attention. We updated the figure. Additionally, line 167 states: **Every cycle consists of a sequence of reproduction steps. Each step selects individuals from the previous generation for a given reproduction pattern. The selected individuals remain in the pool for further reproduction steps per default but can be removed if desired.** To help clarify this further we added on line 266: **Each step is implemented through a pair of selector and offspring factory. The steps are applied sequentially and most of them keep the parent pool unchanged, only one removes the selected parent molecules.**

B. It is unclear what the "main selector" is in the final implementation. Is it selection of top N molecules by ligand efficiency? Please clarify in the text.

We appreciate the reviewer for highlighting the limited description. We added a section at line 172: **The main selector can be freely selected from the available selectors described in Section 2.3.** Additionally, lines 301 and 722 mention that Tournament Selector is used as main selector in the final protocol and for all experiments.

C. It is unclear how many children are spawned at each iteration and how the n+1-th iteration pool is created. Please add this information to the paper. Using the numbers provided in the section 2.6 "Final Protocol", $30+60+30+30+15+30+60 = 255$ molecules are produced by offspring factories at each iteration. This number seems to be constant between generations, then the total number of docked molecules should be $200+255*30=7850$ per run. On line 362 it is mentioned that 20 runs per target were performed, which should result in 157,000 docked molecules. The reported number of docked molecules for each target is given on line 363 and is stated to be between 49000 and 76000 unique molecules. Lines 363-365 state "The difference in sampled molecules per target is due to the stochastic nature of evolutionary optimization, as one run might produce more duplicates than another." Does it mean that around 50%-70% of molecules generated in each run are duplicates? Please clarify how the offspring population is generated and make required alterations to Figure 2 and the text of the Final Protocol. If the above calculations accurately represent the algorithm, please discuss how the duplicates affect the algorithm efficiency.

We would like to thank the reviewer for this very insightful comment. The handling of duplicates is crucial information we did not include in the paper and added it now. To prevent confusion about the deviation in numbers between the reviewers comment and our reply we want to clarify the following. There are only 240 new individuals generated per iteration, as 15 individuals are generated through the identity factory and are just copies. This brings the total number of individuals per run to 7400.

Added to line 160 under algorithm overview: **It should be noted though that whilst we are using the term individual and molecule often together, they are treated differently. Each individual is an entity in the population, participates in the evolutionary optimization process and represents one distinct molecule. However, several individuals can represent the same molecule, as it can happen that several recombinations of parents occur multiple times. This is treated with extra care and explained in more detail in 2.5.**

Added to line 249 under score calculation: **As mentioned before, several individuals can represent the same molecule. This is captured by the algorithm. First, it checks if a molecule has been docked in a previous generation and reuses the same score again. If multiple identical but unscored molecules are present within the same population, only one full docking run is conducted and the same score used for all of the identical copies. To ensure diversity across a population, we apply a similarity penalty. Starting from the best scoring molecule for each molecule a fingerprint is calculated, stored and compared with all stored fingerprints. If the Tanimoto similarity between two fingerprints exceeds 0.95, a penalty of +0.5 is applied to the worse scoring molecule. This is done to prevent a population takeover by a single well scoring molecule. At same time, it does allow very good scoring molecules to be presented by 3-4 individuals (as the first one receives no penalty, the second a penalty of 0.5, the third 1.0, and so on), increasing its chance to pass on genes and thereby pulling the entire population towards a more favorable chemical space.**

Added to line 304 under final protocol: **It is important to make again the distinction between individuals and molecules. The reported numbers are static, meaning that every run results in 7400 individuals. But since these can represent duplicate molecules, we observed on average only 2450-3800 actual molecules being docked per run, as reported in 3.3. While a rate of 50%-70% duplicated molecules seems very high we did not experience a deterioration of results. Additionally, as explained in 2.5, the duplicates do not cause docking overheads and are penalized.**

10. Fig 3, 4, C3: Please adjust the panels for the proteins to be in the same order between the figures for ease of comparison. Make each plot the same size in figures 3 and 4, the empty space can be utilized similarly to C3.

That is an excellent remark. We adjusted the order and resized the two subplots. All three figures are updated.

11. Lines 269-276: Dividing docking score by either the number of heavy atoms or their square root are common strategies to estimate ligand efficiency to correct docking score bias towards heavy molecules. "lid_root2" sounds like a name of a variable from source code and creates confusion. Please replace mentions of "lid_root2" in the paper by "ligand efficiency" or a similar term.

We agree with the reviewer that this is in hindsight a poor naming choice. However, it is already mentioned in two follow up publications and can be found through google scholar. Whilst we would prefer better naming, we suggest keeping it like that to increase the ease of cross-reading.

12. In Figure 5, it is not clear how molecules E and F were recombined into G. The highlights are inconsistent. From the description to the figure, it seems that in molecule E blue highlight should extend to the phenyl ring. In molecule F, there is no obvious way to fragment the molecule such that it would result in a 3-methyl,4-bromo-benzyl group, and the provided highlight is inconsistent. Please revisit the figure and fix this issue.

We updated the figure with new coloring and changed its caption: **Example of the family tree for the best scoring molecule for ABL1 showcasing mutation and crossover. The color highlights do not show synthons, but reappearing structural motifs. A and B recombined through crossover into C, which mutated to D. Here, A includes a tetrazole ring, a common motif in many FDA approved drugs. Both A and B are containing a 1,2,4-triazole, representing the basis-structure to combine it to C. From C to D a mutation changes the position of the oxymethylpyridine at the benzyl-group from 4 to 3. In the other route, E containing a tetrazole and a 1,2,4-triazole exchanges moieties with F containing the substructure 6-Bromo-1,3-benzodioxol. Here, instead of exchanging the exact moiety a similar synthon in the set is searched, adding a 3-methyl,4-bromo-benzyl group to the offspring G. A mutation derivatizes the [1,2,4]triazolo[1,5-a]pyridine from G to H's oxybenzyl-moiety. Finally, both D and H recombine into I, introducing the tetrazole-triazole-system from both H and D while including the pyrazole from D and the 3-methyl,4-bromo-benzyl group from H. All reproduction steps happened in different generations. Positive and negative numbers are observed unfavorable and favorable score changes.**

13. Line 475: "V-SYNTHES uses a greedy heuristic". After carefully looking into V-SYNTHES paper, there was a section called "Structure-guided selection of fragments". Perhaps change the description of heuristic from greedy to structure-based?

The term greedy is added by us to the algorithm and describes the selection criteria for fragments to be built into final molecules. This is a general term for heuristic algorithms within computer science, whereas structure-based, albeit also correct, is exclusive to computer-aided drug discovery. We added a short explanation on line 574 to explain why we labeled V-SYNTHES greedy: **meaning it selects fragments only based on their isolated scores and thereby potentially missing combined molecules which exceed the scores of their building blocks**

14. Line 494: Chemical Space Docking is similar to V-SYNTHES, but not to Deep Docking. This was already discussed in the introduction (lines 65-82).

We apologize for the mix up and replaced Chemical Space Docking with **RosettaVS** (line 595). This tool is similar to Deep Docking and was intended to be mentioned at this position.

15. In appendix, Figure C3 appears without description and seems to be cut off at the bottom.

We would like to thank the reviewer for catching this. We fixed the layout, now the full figure and its description are displayed.

Reviewer #2 comments:

Eisenhuth, Liessman, Moretti and Meiler have developed a computational method called REvoLd based on genetic algorithm that allows users to dock nonenumerated chemical spaces using RosettaLigand which, unlike many other docking tools, is a docking methodology that also takes the protein flexibility into account. They developed the protocol by first docking one million compounds (generated from Enamine REAL Space) to the human dopamine D3 receptor and then optimized the various parameters of the REvoLd using this data so that it retrieves the best docking score as quickly as possible. Finally, the method itself was validated using five different well-known drug targets. It was shown that by docking 49,000-76,000 per target, they were able to retrieve significant amounts of compounds with good RosettaLigand docking scores from the nonenumerated chemical space (Enamine REAL Space). It can be estimated from the runtimes provided in the manuscript that the method requires reasonable computing resources and time as screening one target takes about two days with 1000 cores (one screening consists of 20 parallel runs, which each run consuming 100 cores simultaneously).

The research topic of docking of ultra-large chemical spaces is highly relevant in the context of both academic and industrial drug discovery as demonstrated by the various related studies published recently in the scientific literature and this study fits very well in the scope of Nature Communications Chemistry. The main novelty within this study lies in the use of the computationally expensive RosettaLigand software to screen vast nonenumerated chemical spaces, which might open new avenues for virtual screening of targets where the flexibility of the target is crucial.

The manuscript is well-written, logically constructed and easy to read. The high capability of REvoLd to retrieve good compounds from Enamine REAL Space as scored by RosettaLigand is demonstrated using several targets.

Given the speed of the development in the field of screening of ultra-large chemical spaces has been in last couple of years, it is not surprising to find that some of the most recent related work is not cited and short discussion around those should be added to the manuscript. The few comments below do not require additional experiments, so this reviewer finds the manuscript suitable for publication after these minor revisions:

We thank the reviewer for this kind assessment of our work. We addressed all reviewers' comments carefully and hope that our manuscript is now in line with recent publications in this field.

1) While the main strength of RosettaLigand (and the whole justification for the huge computational demand increase) versus faster rigid docking software (such as GOLD or Glide) should be the consideration of the target protein as a flexible object, the manuscript lacks completely the discussion of benefits for considering protein flexibility in the context virtual screening. This might be obvious to the authors, but this should be thus added to the introduction where this would be discussed in detail and authors should add some concrete examples as well (to quote internet slang: "pics or it didn't happen"). This reviewer is guessing that a lot of development of RosettaLigand has been done since the publication of 2009 paper (<https://www.sciencedirect.com/science/article/pii/S0022283608014289>) and there must be some nice examples of the benefit of using RosettaLigand vs. simple rigid docking.

The reviewer highlights an important point. Therefore we added the following section to line 66: **The majority of these vHTS campaigns utilize rigid docking, as it tremendously decreases the computational demands compared to flexible docking. However, this introduces potential error sources, as rigid docking might not be able to sample favorable protein-ligand structures [7]. This is in line with previous findings where the introduction of both protein and ligand flexibility increased success rates notably [2, 19, 41, 49]. Throughout this study we used the RosettaLigand flexible docking protocol [16, 47, 63]. It is well positioned among other available methods and showed strong ranking capabilities during screens of the Enamine REAL space [29, 42, 63].**

2) Following on that topic: does any of the five targets picked here for benchmarking display any benefits for flexible docking vs rigid docking? This should be discussed around Page 11, line 342. If the targets were not selected because of their flexible nature, the reason why they were selected should be stated clearly. There are plenty targets in DUD-E dataset to choose from so the rationale for picking these specific targets here is interesting to know.

We agree with the reviewer, the reason for selection is unclarified. We added two sentences at line 412 to explain the rationale for selecting these specific targets: **We selected the curated PubChem HTS data to ensure high reliability of reported actives, but only four of the eight reported drug targets had available high quality protein-ligand structures deposited in the PDB. The ABL1 kinase was randomly selected from all DUD-E kinases to increase the diversity of protein classes. Through this, we cover a diverse selection of different valuable drug targets and a broad bandwidth of small-molecule ligands, making them a good test case for the benchmark. Furthermore, they represent a mixture of soluble and membrane proteins, and especially GPCRs are known for their flexible nature.**

3) Section 4.5: The ligand preparation before docking for molecules constructed from Enamine REAL Space needs further elaboration. How does the REvoLd handle stereoisomerism, tautomerism and different

protonation states of the ligands? This is important to describe in detail: for example, the number of different stereoisomers multiply the need for docking calculations as each stereoisomer must be considered as a separate structure to dock. Also, protonation states and tautomers can have a large effect on docking scores and there is no universally accepted approach to deal with them.

We appreciate the reviewer for highlighting this very important point. We used a very basic preparation pipeline which does not consider these differentiations, as we focused our attention on the sampling and optimization of molecules. Therefore we added on line 236: **It should be noted though, that as of now REvoLd is using only the very basic RDKit functionalities for 3D structure generation. For example, there is no special consideration for stereoisomers or different protonation states.**

4) Regarding 3), authors should also elaborate this part there: "RDKit is further used to calculate a three-dimensional embedding of the molecule 208 and a list of low-energy conformers.". There are various ways on generating conformers in RDKit. If they use the ETKDG method, this should be cited as well (<https://pubs.acs.org/doi/10.1021/acs.jcim.5b00654>)

We would like to the reviewer for this remark. We now explicitly state the usage of the ETKDG method and added the corresponding citation at line 234: **RDKit's implementation of the ETKDG method is further used to calculate a three-dimensional embedding of the molecule and a list of low-energy conformers**

5) Section 3.2.: The authors should elaborate the generation of diverse one million compounds a bit more as at least this reviewer is struggling to understand how you select a subset of four reactions from a bigger pool of reactions based on diversity selection of building blocks.

We apologize for not stating this explicitly. We added a section at line 335 to explain the approach: **First, we reduced the number of fragments per reaction and position by selecting a random fragment and adding it to an empty list. Iteratively, all remaining fragments for this specific position calculated a fingerprint-based Tanimoto similarity to all fragments in the list and the one with lowest average similarity got added to the list as well. This is repeated until 500 fragments were added. The entire process is done for all positions and reactions available. Next, we selected a random reaction to start our final set and again iteratively compared the average similarity between all fragments from selected and unselected reactions and added the reaction with the lowest similarity to the selection until it contained four reactions.**

6) Regarding the missing references in the introduction:

a. Page #2, Line #71: When discussing methods that mix conventional docking and machine learning methods, the authors should also mention here MolPal (<https://pubs.rsc.org/en/content/articlelanding/2021/sc/d0sc06805e>) and HASTEN (<https://pubs.acs.org/doi/full/10.1021/acs.jcim.3c01239>).

b. Page #3, Line #77: When discussing methods that approach the problem of screening of ultra-large libraries by docking fragments, SpaceDock should be cited as well (<https://pubs.acs.org/doi/10.1021/acscentsci.3c01521>)

c. Page #3, Line #79: there are also methods that use chemical similarity of whole molecules while still navigating in nonenumerated chemical space and they should be also mentioned here: Thompson Sampling (<https://pubs.acs.org/doi/10.1021/acs.jcim.3c01790>), SpaceGA (<https://pubs.acs.org/doi/10.1021/acs.jcim.4c01308>) and SpaceHASTEN (<https://pubs.acs.org/doi/10.1021/acs.jcim.4c01790>).

We would like to thank the reviewer for bringing these very important and interesting papers to our attention. All mentioned works are added to the manuscript and properly cited. They are discussed on lines:

81 - A similar idea is used for example by Lutten et al. [44], RosettaVS[75], MolPal [27] and HASTEN [61].

85 - SpaceDock follows the same concept, but is not limited to commercially available combinatorial libraries

93 - Other active learning algorithms are SpaceHASTEN and Thompson Sampling

101 - Another approach using an evolutionary (or genetic) algorithm is SpaceGA. It utilizes established mutation and crossover rules and maps the resulting molecules back to the combinatorial chemical space through similarity search with SpaceLight

7) Rosetta requires a separate license for industrial researchers (see <https://els2.comotion.uw.edu/product/rosetta>). The need for this additional license should be explicitly stated in " 5.7 Code availability".

We agree with the reviewer, this is important to be clarified. We added line 685: **Rosetta is available under a permissive license for academic research. Commercial utilization requires a separate license.**